



# Tracking the direct impact of rainfall on groundwater at Mt. Fuji by multiple analyses including microbial DNA

Ayumi Sugiyama[1, 2, 3], Suguru Masuda[1], Kazuyo Nagaosa[1], Maki Tsujimura[3], Kenji Kato[1]

[1] Department of Geosciences, Graduate School of Science, Shizuoka University, Shizuoka 422-8529, Japan
[2] Asano Taiseikiso Engineering Co., Ltd., Tokyo 110-0014, Japan
[3] Faculty of Life and Environmental Sciences, University of Tsukuba, Tsukuba 305-8572, Japan

*Correspondence to*: Kenji Kato (kato.kenji@shizuoka.ac.jp)

**Abstract**. A two to three million tons of spring water flushes out from the foot of Mt. Fuji, the largest volcanic mountain in Japan. Based on the concept of piston flow transport, residence time of stored groundwater at Mt. Fuji was estimated at ~15–
30 years by the $^{36}Cl/Cl$ ratio (Tosaki et al., 2011). This range, however, represents the average residence time of groundwater that was mixed before it flushed out. To elucidate the route of groundwater in a given system, we determined signatures of direct impacts of rainfall on groundwater, using microbial, and stable isotopic (delta $^{18}O$), and chemical analyses (concentration of silica). Chemical analysis of the groundwater gave an average value of the water, which was already mixed with waters from various sources and routes in the subsurface environment. The microbial analysis suggested locations of water origin and
paths.

In situ observation during four rainfall events revealed that the stable oxygen isotopic signature obtained from spring water (at 726 m a.s.l., site SP-0m) and shallow groundwater (at 150 m a.s.l., site GW-42m), where the average recharge height from rainfall was 1500–1800 m, became greater than values observed prior to a torrential rain producing more than 300 mm of precipitation. The concentration of silica decreased after this event. In addition, the abundance of *Bacteria* in spring water
increased, suggesting the influence of the heavy rain. Such changes did not appear when rainfall was less than 100 mm per event. The above findings indicate a rapid flow of rain through the shallow part of the aquifer, which appeared within a few weeks of the torrential rain extracting abundant microbes from soil in the studied geologic setting. Interestingly, we found that after the torrential rain, the abundance of *Archaea* increased in the deep groundwater at site GW-550m, ~12 km downstream of SP-0m. However, chemical parameters did not show any change after the event. This suggests that strengthened piston flow
caused by the heavy rain transported archaeal particles from the geologic layer along the groundwater route. This finding was supported by changes in constituents of *Archaea*, dominated by *Halobacteriales* and *Methanobacteriales*, which were not seen from other observations. Those two groups of *Archaea* are believed to be relatively tightly embedded in the geologic layer and were extracted from the environment to the examined groundwater through enforced piston flow. Microbial DNA can thus give information about the groundwater route, which may not be shown by analysis of chemical materials dissolved in the
groundwater.



# 1 Introduction

Though runoff process of stream water and runoff peak response time of streams influenced by rainfall have been well studied (e.g., Hubert et al., 1969; Onda et al., 1999; Asai et al., 2001; Tekleab et al., 2014), runoff processes of groundwater affected directly by rainfall is not precisely explained. The contribution of rainfall through subsurface pass to stream water

was estimated by preceding studies, but they did not address the route of groundwater until it affected streamflow.

To get indication on the route of groundwater we herein newly applied microbial DNA analysis focusing on heavy rainfall at the foot of Mt. Fuji located in central Japan, which is the largest Quaternary stratovolcano in Japan with a peak at 3,776 m a.s.l. At the foot of this mountain we previously found that pH of groundwater decreased from 7.29 to 7.02 a few weeks after a typhoon in August and September 2011 (total rainfall was > 800 mm) (Segawa et al., 2015) at 200 m a.s.l. This decrease of

pH was probably influenced by low pH of the rainwater (pH from 4.7 to 6.4; Watanabe et al., 2006). This rapid decrease of pH cannot be explained by piston flow transport of groundwater in which newly supplied water pushes out older water preserved in the subsurface bed (e.g., Bethke and Johnson, 2008). Considering the pH of rainfall at Mt. Fuji, the lowering of groundwater pH suggested that the newly supplied rainwater mixed directly with groundwater over a period of weeks.

In addition, our preceding microbiological study of groundwater at the foot of Mt. Fuji furnished a clue to estimate possible

groundwater routes by finding thermophilic bacterial DNA in spring water, whose temperature was as low as ~10–15 °C throughout the year (Segawa et al., 2015). Thermophilic prokaryotes are optimally adapted to temperatures > 40 °C. This suggests that at least some of the groundwater source was at a depth of 600 m or greater, based on a temperature gradient of 4 °C/100 m. This depth is far below the lava layer that was taken to be a substantial pool of this groundwater (Tsuchi, 2007). Thus, microbial information can help estimate the route of groundwater.

Following the above findings, we tried to estimate the groundwater route with a focus on heavy rainfall, by tracing the signature of direct rainfall impacts. This was done using (i) stable oxygen and hydrogen isotopic analysis to track the movement of water molecules, (ii) chemical analysis of silica concentration in groundwater, which indicates its possible dilution by rainwater with low silica concentration, and (iii) microbial analysis including DNA sequencing to estimate the groundwater route, which may include a possible function of microbes in a given geological environment. Whereas stable isotopic and

chemical analyses show average values of the water originated from various sources, microbes transported by groundwater suggest the route and place where they proliferated through their eco-physiological characteristics constrained by their optimal growth condition. In other word, microbial DNA brings a message of their route, though the examined water was already blended with various groundwater prior to examination. To elucidate microbial properties in the studied groundwater, we used total direct counting (TDC) of prokaryotes, catalyzed reporter deposition – fluorescence in situ hybridization (CARD-FISH),

16S rRNA gene-targeted polymerase chain reaction (PCR), denaturing gradient gel electrophoresis (DGGE), and a next-generation sequencing. Here, we first employed microbial analysis to reveal the groundwater route in the shallow and deep subsurface environment.



## 2 Materials and methods

### 2.1 Study site

The study area is at the foot of Mt. Fuji, in the central part of the main island of Japan. Two geologic formations are present (Tsuya, 1968). The upper formation (lava flow and tephra) is permeable and the lower one (mudflow) is impermeable or has

low permeability. According to a water flow simulation (using the General Purpose Terrestrial Fluid – FLOWS simulator or GETFLOWS, Geosphere Environmental Technology Co., Ltd.; Kato et al., 2015; Fig. 1), one of the groundwater bodies at the foot of Mt. Fuji flows toward Suruga Bay together with the Urui River through the western lava flows. We selected this flow as a target study location. Groundwater was sampled at sites SP-0m (Shibakawa, 726 m a.s.l., from spring water), GW-42m (Yodoshi, 150 m a.s.l., from shallow well water at 42 m depth), and GW-550m (Aoki, 175 m a.s.l., from deep well water at

550 m depth). An unusual flush out of groundwater was observed at GW-42m after heavy rainfall from a typhoon in September 2011. Horizontal distance between SP-0m and GW-42m was ~12 km. GW-42m and GW-550m are at a distance of 0.2 km. Rainwater was sampled at five sites with varying altitudes (R1 at 2,364 m a.s.l., R2 at 1,431 m a.s.l., R3 at 1,081 m a.s.l., R4 at 850 m a.s.l., and R5 at 723 m a.s.l.).

### 2.2 Field measurements and sample collection

Simultaneous observation of rainwater and groundwater was done from June 2012 to November 2014. Spring water was sampled again for gene analysis on 10 March 2015; rainfall was not observed for a few weeks prior to this date. Physical and chemical parameters of groundwater and rainwater were measured directly at the sampling site. In situ measurements of water temperature, pH, electric conductivity (EC), dissolved oxygen (DO) and oxidation reduction potential (ORP) of groundwater were made using an IM-22P (DKK-TOA Co., Tokyo, Japan), CM-14P (DKK-TOA Co.), Hach DO meter (Hach Co., Loveland,

CO, USA), and RM-20P (DKK-TOA Co.). ORP values were transformed to $E_h$ values (Pt conversion) by Equation (1). Water temperature, pH and EC were measured for rainwater.

$$E_h \text{ (Pt) [mV]} = \text{ORP [mV]} + (-0.7198 \times \text{Temperature [°C]} + 224.36) \qquad (1)$$

Groundwater was collected directly into sterilized bottles or using a sterilized jug.

Rainwater for isotopic, chemical and microbial analysis was collected using 500 mL and 1,000 mL Pyrex bottles with 300

mm and 210 mm funnels, respectively (Hamers et al., 2001; Beiderwieden et al., 2005). Rainwater samplers prevented from evaporation using a plastic ball in between the bottom of funnel and glass bottle were set up at 1.5 meters above ground. To measure silica concentration, rainwater was collected using a plastic jug. The samples were immediately kept cool in the dark until treatment.



## 2.3 Isotopic and chemical analysis

Samples for analysis of stable isotopes ($^{18}O$ and D) and major ions ($L^+$, $Na^+$, $NH_4^+$, $K^+$, $Mg^{2+}$, $Ca^{2+}$, $F^-$, $Cl^-$, $NO_2^-$, $Br^-$, $SO_4^{2-}$, $PO_4^{3-}$, $NO_3^-$ and $CO_3^{2-}$) were filtered through a 0.22-µm Millex-GS filter (EMD Millipore, Billerica, MA, USA) and stored at 4 °C and −20 °C, respectively. Samples for measurement of silica concentration were stored at 4 °C until analysis.

Stable isotope ratios of oxygen and hydrogen ($\delta^{18}O$, $\delta D$) were measured with a water isotope analyzer (Picarro L2130-i, L2120-i; Picarro Inc., Santa Clara, CA, USA). The isotopic ratio was calculated from the ring-down time at specific wavelengths, using wavelength-scanned, cavity ring-down spectroscopy technology (Gupta et al., 2009). The $\delta^{18}O$ and $\delta D$ values are expressed in per mil (‰) after normalization to the Vienna Standard Mean Ocean Water scale. Analytical errors for $\delta^{18}O$ and $\delta D$ were 0.025‰ and 0.1‰, respectively.

Concentrations of major dissolved ions were measured by ion chromatograph (ICS-3000, ICS-2100; Thermo Fisher Scientific Inc., Waltham, MA, USA). Dissolved silica concentration was determined by the molybdenum yellow colorimetric method using GeneQuant 100 (GE Healthcare UK Ltd., Buckinghamshire, England).

## 2.4 Microbial analysis

### 2.4.1 Microscopic analysis

TDC was conducted after Porter and Feig (1980) with some modifications, to elucidate the abundance of prokaryotes. CARD-FISH was performed after Pernthaler et al. (2002) and Teira et al. (2004) to analyze microbial community structure. Samples for TDC were fixed in pH-neutral formaldehyde (Wako Pure Chemical Industries Ltd., Osaka, Japan). A subsample of groundwater and rainwater was collected on a Nuclepore filter (0.2 µm pore size; GE Healthcare UK Ltd.). Every treatment of groundwater samples was done on a clean bench (AS ONE Co., Osaka, Japan). Cells were stained with 4′,6-diamidino-2-
phenylindole (DAPI, Nacalai Tesque Inc., Kyoto, Japan; final concentration, 0.01 µg $mL^{-1}$) and more than 500 prokaryotic cells were counted under epifluorescence microscopy (BX51-FLA; Olympus Corp., Tokyo, Japan). Samples for CARD-FISH were fixed in paraformaldehyde (final concentration 3%; Wako Pure Chemical Industries Ltd.). Hybridization was performed at 35 °C for 2 h using a horseradish-peroxidase (HRP) labeled *Bacteria*-specific Eub338 probe (5′-GCT GCC TCC CGT AGG AGT-3′; Amann et al., 1990), or 12 h using a HRP-labeled *Archaea*-specific ARC915 probe (5′-GTG CTC CCC CGC CAA
TTC CT-3′; Stahl and Amann, 1991) or Non915 probe (5′-TCC TTA ACC GCC CCC TCG TG-3′). Numbers of *Bacteria* and *Archaea* were counted based on pictures taken under epifluorescence microscopy (BX51-FLA; Olympus) equipped with a digital camera (DP71, Olympus).

### 2.4.2 DNA extraction

Extraction, amplification and sequencing of DNA were done according to the method of Kimura et al. (2007). A 10-L sample
of groundwater was filtered using a 0.22-µm Sterivex-GV filter (EMD Millipore). A ~1.5-L rainwater sample was filtered using the same filter. Bulk DNA was extracted using the method described by Somerville et al. (1989). Prokaryotic cells were





lysed with a solution of lysozyme and proteinase K in Sterivex-GV. Bulk DNA was extracted with phenol. Concentration and purity of extracted DNA were checked by a NanoVue GE Healthcare UK Ltd.).

### 2.4.3 16S rRNA gene amplicon sequencing

The 16S rRNA genes were amplified by PCR using the primers coded V3-V4 regions Pro341F (5′-TCG TCG GCA GCG TCA
GAT GTG TAT AAG AGA CAG CCT ACG GGN BGC ASC AG-3′) / Pro805R (5′-GTC TCG TGG GCT CGG AGA TGT GTA TAA GAG ACA GGA CTA CNV GGG TAT CTA ATC C-3′) primer set (Claesson et al., 2010; Klindworth et al., 2012; Takahashi et al., 2014). The following reaction mixture (total volume 25 µL) was used: 12.5 µL 2×buffer, 0.5 µL MightyAmp (Takara Bio Inc., Otsu, Japan), and 5 pmol of each primer. The following PCR conditions were used: an initial denaturation step at 98 °C for 2 min followed by 15 cycles of denaturation at 98 °C for 10 s, annealing at 55 °C for 15 s, and extension at
68 °C for 30 s. Amplification was completed with 5 min at 68 °C. PCR products were purified using the SPRIselect kit (Beckman Coulter, Brea, CA, USA). Sample libraries for sequencing were prepared according to the MiSeq™ Reagent Kit Preparation Guide (Illumina Inc., San Diego, CA, USA), and next generation sequencing was run using MiSeq. Operational taxonomic units (OTUs) were clustered using Macqiime (v1.9.1) at a 97 % similarity level. Chimeric sequences were detected by Macqiime via ChimeraSlayer. Individual OTUs were assigned based on representative sequences using Classifier from
Ribosomal Database Project II (Wang et al., 2007) with Green Genes database (http://greengenes.lbl.gov/). Sequence data were submitted to the DNA Data Bank of Japan, accession number DRA004571.

## 3 Results

### 3.1 Rainfall and environmental parameters

We studied four rainfall events from 2012 to 2014 at the foot of Mt. Fuji. Annual precipitation in the study area was 1,971 mm
in 2013 and 2,403 mm in 2014. In event 1, 17 October 2012, there was 30 mm of rain. In event 2, 4–5 September 2013 and 15–16 September 2013 (typhoon), precipitation exceeded 300 mm. In event 3, 10 July 2014, precipitation was 100 mm. In event 4, 5–6 October 2014 and 13–14 October 2014, precipitation was > 300 mm (Fig. 2a).

Measured environmental parameters of rainwater and groundwater are summarized in Table 1. Temperatures and pH of rainwater ranged from 10.1 to 31.9 °C and 3.94 to 6.29, respectively. The observed range of pH was slightly acidic.
Temperatures of groundwater at SP-0m, GW-42m and GW-550m were 10.1–20.6°C. pH, EC and Eh (Pt) (Table 1) of groundwater at those three sites were 5.84–8.26, 49.5–161.5 µS cm$^{-1}$ and 246–497 mV, respectively. The degree of saturation with respect with DO concentration of groundwater was 73.0%−100.7%, suggesting that it was nearly saturated or oversaturated.

There was a remarkable rise in temperature and decrease in pH after a few weeks of event 2 in spring water at SP-0m,
which suggests a direct influence of torrential rainfall on groundwater.





### 3.2 Stable isotopic analysis of rainwater and groundwater

The oxygen isotopic ratio of rainwater was −6.35‰ for event 1, −8.63‰ to −6.63‰ for event 2, −11.52‰ to −9.19‰ for event 3, and −8.34‰ to −4.68‰ for event 4. That ratio of $\delta^{18}O$ in groundwater was −11.36‰ to −8.63‰ (n=46). The hydrogen isotopic ratio of rainwater and groundwater was −95.19‰ to −13.42‰ and −76.13‰ to −57.31‰, respectively.

The oxygen isotopic ratio measured in deep (550-m depth) groundwater at GW-550m was −11.36‰ to −10.81‰ (n=13), being smaller than that of spring water at SP-0m (−9.38‰ to −8.63‰; n=20) and shallow (42-m depth) groundwater at GW-42m (−9.26‰ to −8.69‰; n=13). The altitude effect of rainwater was calculated from observation at −0.25‰ per 100 m with respect to the oxygen isotopic ratio and −2.35‰ per 100 m with respect to the hydrogen isotopic ratio. From those numbers, the major supply of rainwater was expected to be 1,700 m a.s.l. for the spring water and groundwater at SP-0m and GW-42m, whereas it was 2,500 m a.s.l. for GW-550m from the recharge-water line of Mt. Fuji (Yasuhara et al., 2007).

The stable isotope signature of groundwater showed an increase in $\delta^{18}O$ at SP-0m after events 2 and 4, from −9.32‰ to −8.83‰ and −8.79 to −8.63‰, respectively, with precipitation exceeding 300 mm (Fig. 2b). After those events, $\delta^{18}O$ decreased to the level observed before the event. A similar tendency was observed in shallow groundwater after event 2 at GW-42m, ~12 km downstream of SP-0m (Fig. 2c). However, such change was not observed in deep groundwater at GW-550m, located near GW-42m (Fig. 2d). In contrast with the heavy rainfall, events 1 and 3 with precipitation less than 30 and 100 mm, respectively, did not show increased $\delta^{18}O$ at SP-0m, GW-42m and GW-550m, except for a slight increase at SP-0m after event 3. For the heavy rainfall of event 4, we observed a weak but gradual increase in $\delta^{18}O$ and subsequent decrease at SP-0m, but this was not evidenced at GW-42m. The average $\delta^{18}O$ prior to event 4 was larger than that preceding event 2. This might have caused a weak influence of rainfall on the groundwater, because that influence was expected to have already accumulated in the groundwater.

### 3.3 Chemical analysis of rainwater and groundwater

Although the silica concentration of rainwater was 0.0–3.1 mg $L^{-1}$ (n=48), that of groundwater was higher, 24.6–46.6 mg $L^{-1}$ (n=45). At SP-0m, the torrential rainfall of event 2 decreased the silica concentration in groundwater following the event (Fig. 2b). There was a similar finding at GW-42m, though such change was not observed in deep groundwater at GW-550m. There was no apparent decline in silica concentration after events 1 and 3, even at SP-0m. The heavy rainfall of event 4 also did not cause a subsequent decrease in silica concentration.

Most groundwater so far examined was categorized as of the Ca-HCO$_3$ type with their significant concentration, while GW-550m was different from them due to its high concentration of Na$^+$ (supplemental information, Fig. S1). The concentration of analyzed ions in rainwater was very small.



### 3.4 Microbial analysis of rainwater and groundwater

### 3.4.1 Abundance of prokaryotes

Abundance of prokaryotes of rainwater ranged from $(4.04 \pm 0.02) \times 10^4$ to $(1.67 \pm 0.08) \times 10^6$ cells mL$^{-1}$ (n=48), which significantly exceeded that of groundwater, whose range was $(6.86 \pm 1.53) \times 10^2$ to $(1.12 \pm 0.09) \times 10^4$ cells mL$^{-1}$ (n=45). The

abundance of prokaryote in groundwater was two orders of magnitude smaller than that of rainwater. CARD-FISH revealed *Bacteria* and *Archaea* respectively comprised 20.7% to 40.0% (n=8) and 0.8% to 6.6% (n=8) of the total number of prokaryotes in rainwater. Under such ordinary low abundance of prokaryotes in groundwater, there was an apparent influence in event 2 at SP-0m. SP-0m was located ~1 km below the average altitude of the recharge zone. Abundance of *Bacteria* at SP-0m increased sharply after event 2, from $2.6 \times 10^2$ cells mL$^{-1}$ to $1.7 \times 10^3$ cells mL$^{-1}$, and total abundance of prokaryotes

increased from $3.21 \times 10^3$ cells mL$^{-1}$ to $1.12 \times 10^4$ cells mL$^{-1}$ (Fig. 3a). A similar phenomenon did not appear at GW-42m, in shallow groundwater flushed out ~12 km downstream of SP-0m (Fig. 3b). In addition, there was a very interesting increase in the abundance of *Archaea* in deep groundwater at GW-550m, 12 km downstream of SP-0m, where water was obtained from 550-m depth. The number of *Archaea* increased from $3.0 \times 10^1$ cells mL$^{-1}$ to $1.9 \times 10^2$ cells mL$^{-1}$ at GW-550m (Fig. 3c). A similar phenomenon was observed after event 4, though the response was somewhat weaker than in event 2.

### 3.4.2 Bacterial community constituents

Next-generation sequencing retrieved diversified community constituents at the level of order with 384, 268 and 278 from rainwater (R5), spring water before event 2 (SP-0m-1) and spring water after event 2 (SP-0m-2), respectively. Five groups with orders *Burkholderiales*, *Rhizobiales*, *Sphingobacteriales, Pseudomonadales* and *Sphingomonadales* comprised about 90% of all constituents of the R5 community (Fig. 4). In contrast to rainwater, for spring water before event 2 (SP-0m-1), only

*Burkholderiales* were major constituents among the five major groups retrieved from rainwater, with 17.3% of the entire bacterial community. The bacterial community in spring water (SP-0m-1) was more diversified than that in rainwater, R5. A clear difference in bacterial community structure appeared at SP-0m after torrential rainfall (SP-0m-2 in Fig. 4). After event 2, *Burkholderiales* became apparently dominant in groundwater at SP-0m (58.8%). Following *Burkholderiales*, *Flavobacteriales* was substantial with 8.9%, but had a contribution that did not vary before and after the event. *Rhizobiales*, *Sphingobacteriales,*

*Pseudomonadales* and *Sphingomonadales*, which were the major dominant groups in rainwater following *Burkholderiales*, were apparently not retrieved from the spring water at SP-0m after event 2. In contrast, *Bdellovibrionales* and *Bacillales*, which were not a major constituent at SP-0m before that event, became dominant after the torrential rainfall.

### 3.4.3 Archaeal community constituents

We examined the archaeal community, focusing on deep groundwater at GW-550m, where a remarkable increase was observed

in the abundance of *Archaea* after event 2. Next-generation sequencing retrieved 12 major groups of *Archaea* at the level of order for each analyzed sample. Rainwater (R5 in Fig. 5) comprised WCHD3-30 and YLA114 belonging to *Parvarchaeota*,





E2 belonging to *Thermoplasmata*, unclassified *Euryarchaeota, Cenarchaeales*, unclassified MBGA (*Crenarchaeota*), and *Micrarchaeles.* WCHD3-30 and YLA114 also dominated in deep groundwater of GW-550m-1 in the non-rainy period, followed by E2 and *Cenarchaeales*. The relative proportion of each order group did not vary much after event 4 (GW-550m-3) and the non-rainy period (GW-550m-1). However, a few weeks following the torrential rainfall of event 2 (GW-550m-2), *Halobacteriales* and *Methanobacteriales* were predominant in the deep groundwater. These relative constituents to the whole community were clearly different from other results.

## 4 Discussion

Tracer hydrology studies of rainfall-runoff processes have revealed the mixing process of rainfall and groundwater in stream water shortly after heavy rain in a range from days to weeks (e.g., Pearce et al., 1986; McDonnell et al., 1991; Silliman and Booth, 1993; Blume et al., 2008). A sharp decrease in pH of spring water influenced by heavy rainfall, suggesting a direct effect of rainwater on groundwater, was observed at the foot of Mt. Fuji (Segawa et al., 2015). Following these studies, we investigated heavy (> 300 mm) and light (100 mm) rain at the foot of volcanic Mt. Fuji at sites SP-0m and GW-42m (shallow groundwater) and GW-550m (deep groundwater), where average recharge of rainfall and snowfall was estimated between 1,700 and 2,500 m a.s.l.

We found fast flow of groundwater caused by torrential typhoon rainfall in multiple analyses, including those of microbes. Rainwater exceeding 300 mm traversed the very shallow portion of the subsurface aquifer and appeared 2 weeks after the event at SP-0m. That site is ~1 km lower in altitude and ~5–7 km downstream horizontally from the average recharge area of the rainfall. This finding was deduced from the movement of microbial particles and of water molecules tracked by the stable isotope signature of $\delta^{18}O$, as well as measurement of dissolved silica concentration. The silica concentration in groundwater is ascribable to the extraction of silicate from soil and rock (Wels et al., 1991; Asano et al., 2003). Thus, decrease of that concentration in groundwater after torrential rain suggests that the flow of groundwater was substantially faster than usual, or a dilution of the concentration by great amounts of infiltrated water. Rapid flow of groundwater was also detected in shallow groundwater at GW-42m, which was ~600 m below the altitude of SP-0m and ~12 km downstream. This finding was associated with an increase in $\delta^{18}O$ and decline in silica concentration.

The effect of torrential rainfall was also clearly detected by a sharp increase in the abundance of *Bacteria* at site SP-0m. An apparent predominance in the bacterial community of *Burkholderiales* suggests incorporation of microbes from soil through extraction by enforced flow rate, because the abundance of prokaryotes in soil is about four or five orders of magnitude greater than that of groundwater (reviewed by Whitman et al., 1998), and *Burkholderiales* is known to inhabit the soil environment (Garrity et al., 2005a). Direct incorporation of microbes from rainwater (Fig. 4, R5 and SP-0m-2) is another possibility. Additional analysis with DGGE showed that *Herbaspirillum* sp. belonging to *Burkholderiales* was retrieved from SP-0m spring water 2 weeks and 5 days after event 2 (DGGE, supplemental information Table S1). Infiltration of microbes from the soil matrix, however, seems more likely, because the second, third and fourth dominant groups of *Bacteria* in



rainwater, *Rhizobiales*, *Sphingomonadales* and *Pseudomonadales* were not significantly retrieved from the spring water after event 2. Such extraction might increase the relative abundance of *Bdellovibrionales* in groundwater, a group known to generally inhabit the soil environment (Garrity et al., 2005b).

Furthermore, sequences affiliated with thermophilic bacteria was scarcely retrieved from the samples of the examined SP-0m after event 2, which supports the assertion of enforced piston flow through a deep subsurface zone > 600 m which given temperature exceeding 40 °C, where thermophilic bacteria inhabited was not considerable. Viral particles have previously been used as a tracer of water movement. Hunt et al. (2014) showed preferential flow paths using this method. Viral particles only provide information on groundwater flow paths, whereas microbial analysis including DNA provides additional information on the location of origin of the microbes and the magnitude of impact of water flow on microbes extracted from geologic layers. Thus, microbial analysis can give insight into the route of groundwater through both shallow and deep environments. The latter is discussed below.

In contrast to the findings for shallow groundwater and spring water, no direct influence of torrential rainfall was detected in either the stable isotope signature or concentration of silica in deep groundwater at GW-550m (~12 km downstream of SP-0m) after event 2. Considering the difference of horizontal distance and depth from which water was sampled between GW-42m and GW-550m, the direct impact of rainfall from the observation is expected to be barely noticeable at 550 m depth.

However, we observed an interesting increase in abundance of *Archaea* at GW-550m 2 weeks after event 2, which was supported by an apparent change in constituents of archaeal OTUs. *Halobacteriales*, which inhabit environments with high concentrations of sodium and *Methanobacteriales*, a strict anoxic methane producer, were dominant members after the torrential rainfall. It has been shown that archaeal abundance increased with depth in both terrestrial (Kato et al., 2009) and marine (Lipp et al., 2008, Inagaki et al., 2015) subsurface environments. Deep groundwater in the study area contained high concentrations of $Na^+$, from 14.3 mg $L^{-1}$ to 14.6 mg $L^{-1}$ (n=13), while these were 5.4 mg $L^{-1}$ to 7.8 mg $L^{-1}$ (n=32) in groundwater at SP-0m and GW-42m (supplemental information Fig. S1). Thus, not only strict anaerobic but halophilic archaea may be abundant within the deep subsurface environment of the study area, although they were not retrieved from groundwater in other examinations, likely because they were embedded in the matrix of geologic layers. An augmented flow rate caused by torrential rainfall might have extracted *Halobacteriales* and *Methanobacteriales* from the matrix of those layers into the studied groundwater (Fig. 5). In contrast to the spring water (SP-0m), some sequences affiliated with thermophilic bacteria were retrieved from the deep groundwater (GW-550m), which suggested microbes in the deep groundwater contained *in situ* populations. This suggests that strengthened piston flow caused by the heavy rain transported archaeal particles from the deep geologic layer along the groundwater route.

A possible reason why there was no apparent influence of heavy rain on microbial particles in groundwater at GW-42m, ~12 km downstream of SP-0m, may be attributable to the trapping of microbial particles by soil and lava across the flow trajectory. There was a question whether in situ population change through the growth in groundwater could be explained by their estimated growth rates. The doubling time of prokaryotes in the groundwater was estimated at 85 days, from the observed frequency of dividing cells' to the entire population (Newell and Christian, 1981). Thus, the possibility of altered populations




via growth within a few weeks may be small. Microbes observed in the groundwater may represent the original locations where they grew.

Wels et al. (1991) separated streamflow into three components, surface water, soil water and groundwater, using a two-step separation with stable isotopic ratios and silica concentration. Based on their method, an estimated 21% and 5% of water in the subsurface environment at SP-0m and GW-42m, respectively, were attributed to soil water following event 2. The effect of soil water at SP-0m was estimated to be stronger than that at GW-42m. This difference appeared in microbial abundance and constituents as well as in water molecule movement. This supports the sudden appearance of rapid flow through the shallow aquifer consisting partly of soil layers in groundwater at SP-0m. This phenomenon was driven by heavy rainfall.

In contrast, direct and rapid effects of rainwater movement into groundwater were not observed for weak rainfall, as evidenced by the results of events 1 and 3. However, a question remains as to why an impact similar to event 2 was not observed for another heavy rainfall, that of event 4. The impact of rainfall before event 4 persisted, as suggested by a stronger $\delta^{18}O$ signature, which might have veiled that impact (Fig. 2).

In addition to the chemical analyses of groundwater, we showed that microbes could show the route of groundwater in the invisible subsurface environment.

## 5 Conclusions

Chemical analyses using stable isotopes and dissolved ions show the properties of the groundwater mixed throughout the route it flowed. In contrast, microbial particles suggest the locations where they were incorporated in the groundwater as far as they survived. Thus, microbial analysis can provide information about the origin and route of the groundwater.

Next-generation gene sequencing provides detailed information of high resolution on the examined microbial community constituents, which was never attained by the conventional gene sequencing technique targeting small subunit ribosomal DNA. Previous conventional gene sequencing does not give quantitative information on microbial community constituents. Here, we first indicated the route of groundwater using a next-generation sequencing analysis of *Bacteria* and *Archaea*. Bacterial abundance and community constituents showed that torrential rainfall caused rapid flow of rainwater through the shallow part of the aquifer, and archaeal abundance and constituents suggested fast and accelerated piston flow in deep groundwater within a few weeks after that rainfall. The former finding was mostly ascertained by the chemical analysis, but the latter finding was not shown by chemical analysis.

*Acknowledgments.* This study was supported by a Grant-in-Aid for Scientific Research from the Japan Society for the Promotion of Science (JSPS KAKENHI Grant Number JP26257402) and a research fund of the River Front Research Institute. The work was done with support from a Joint Research Grant for Environmental Isotope Study of the Research Institute for Humanity and Nature. 16S rRNA gene amplicon sequencing was carried out at the Functional Genomics Section, Research Institute of Green Science and Technology, Shizuoka University.





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





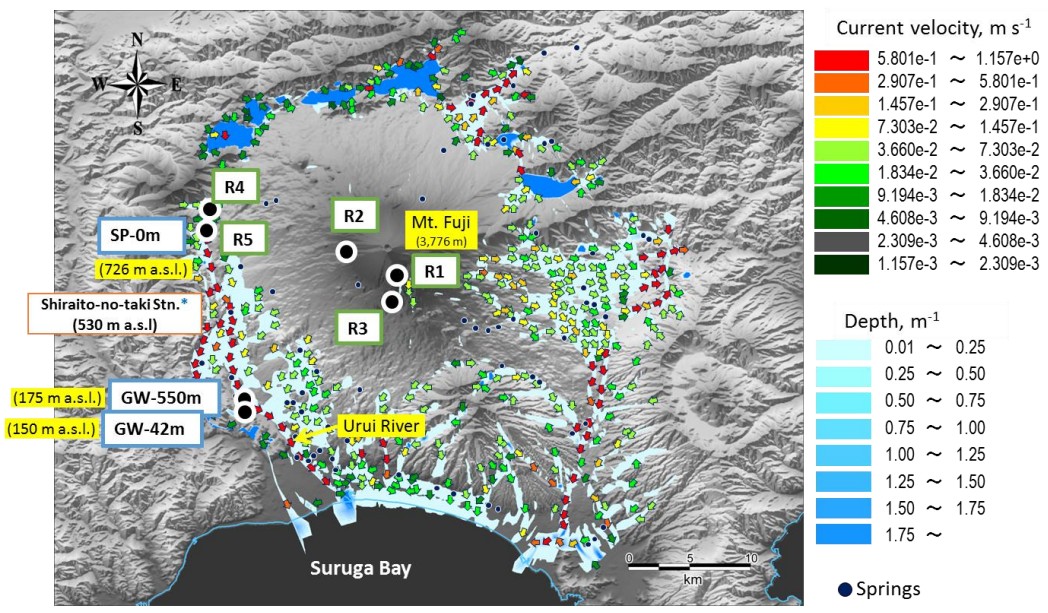

Figure 1. Study sites in western foot of Mt. Fuji. Red arrows indicate main fast flow (GETFLOWS; Kato *et al.*, 2015 partially modified). Precipitation is sampled at R1 to R5. Groundwater is sampled at SP-0 m, GW-42 m and GW-550 m. R1 at 2,364 m a.s.l., R2 at 1,431 m a.s.l., R3 at 1,081 m a.s.l., R4 at 850 m a.s.l. and R5 at 723 m a.s.l. SP-0 m, Shibakawa at 726 m a.s.l., spring water, GW-42 m, Yodoshi at 150 m a.s.l., shallow well water of 42 m, GW-550 m, Aoki at 175 m a.s.l., deep well water of 550 m. * Amount of precipitation for the studied area was recorded at Shiraito-no-taki Station of Japan Weather Association.




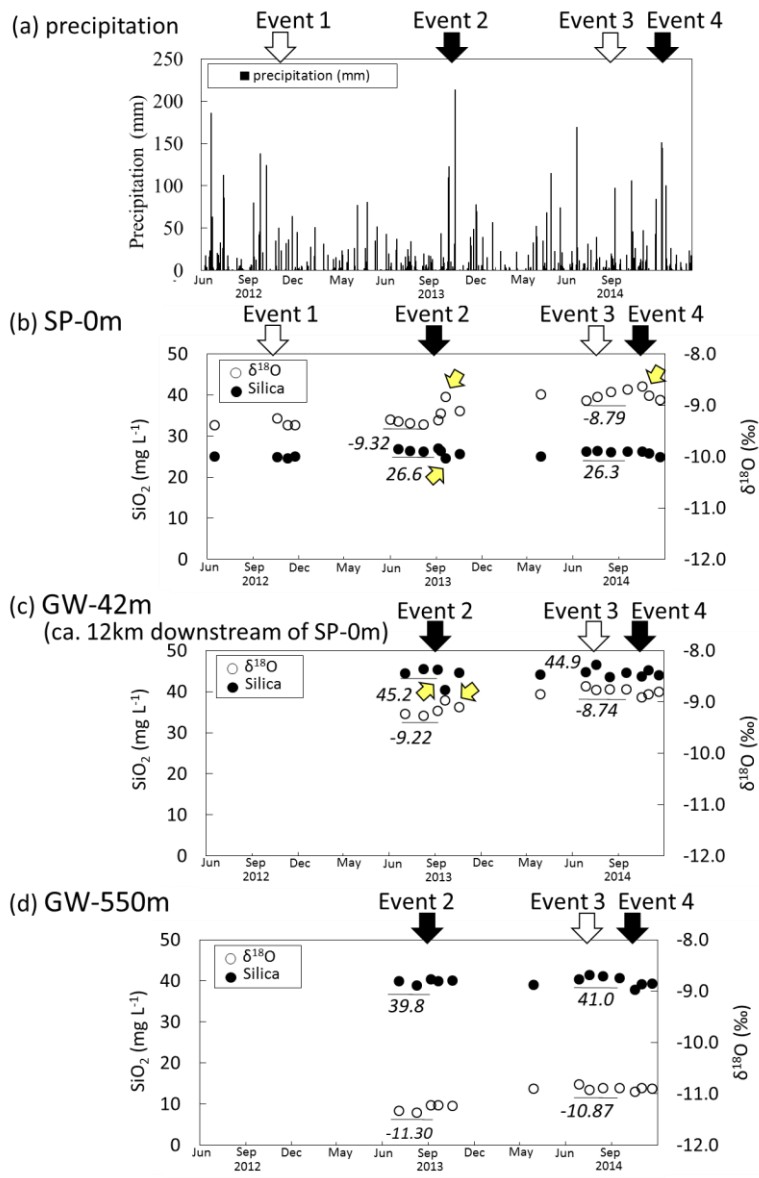

Figure 2. Changes in oxygen isotope ratio, concentration of silica. (a) Precipitation, (b) Groundwater at SP-0m, (c) shallow groundwater at GW-42m, (d) deep groundwater at GW-550m. Black and open arrows indicate the rainfall event; Event 1, Event 2, Event 3 and Event 4. Black arrows particularly indicate the torrential rainfall. Yellow arrows indicate the signature of direct impact of rainfall. The number shows an average value of the plots underline.





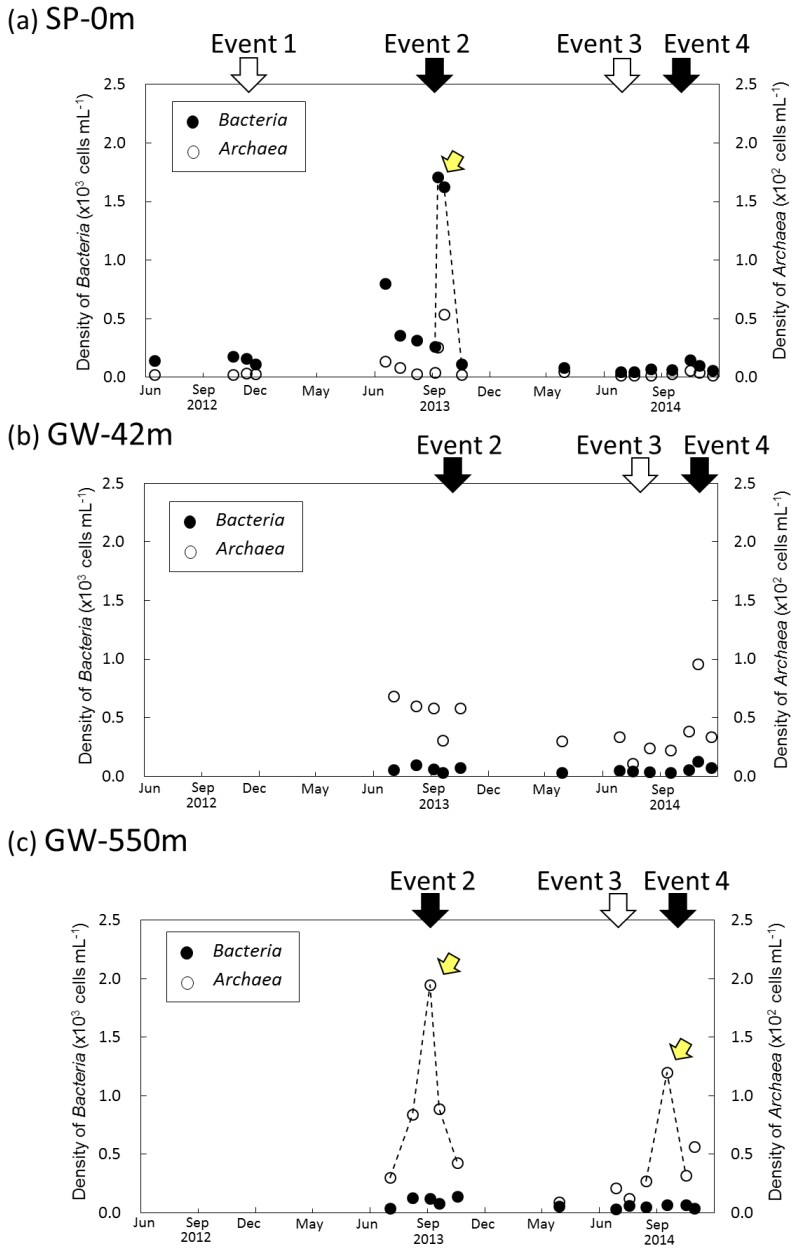

Figure 3. Changes in community structure of prokaryotes in groundwater. (a) Groundwater at SP-0m, (b) shallow groundwater at GW-42M, (c) deep groundwater at GW-550m.  Black and open arrows indicate the rainfall event; Event 1, Event 2, Event 3 and Event 4. Black arrows particularly indicate the torrential rainfall. Yellow arrows indicate the signature of impact of rainfall.



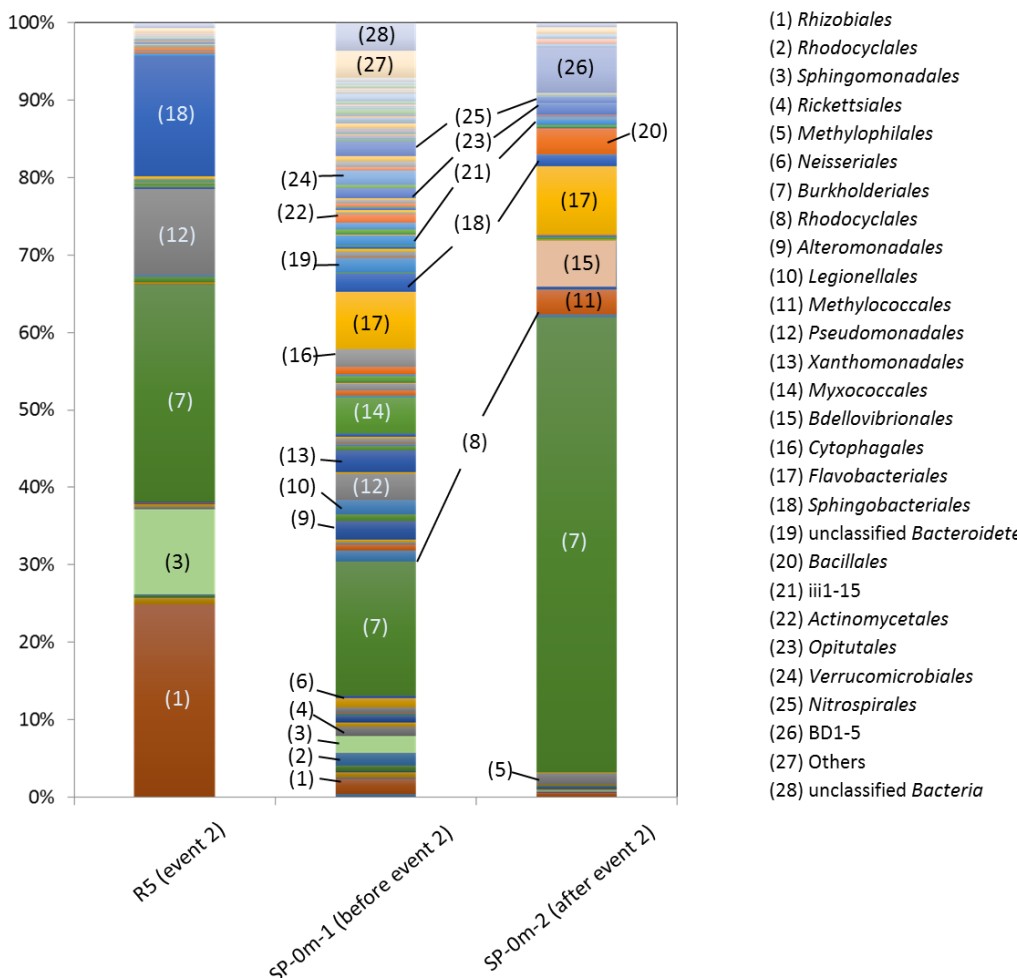

Figure 4. Relative abundance of bacterial orders in each events by next-generation sequencing analysis.



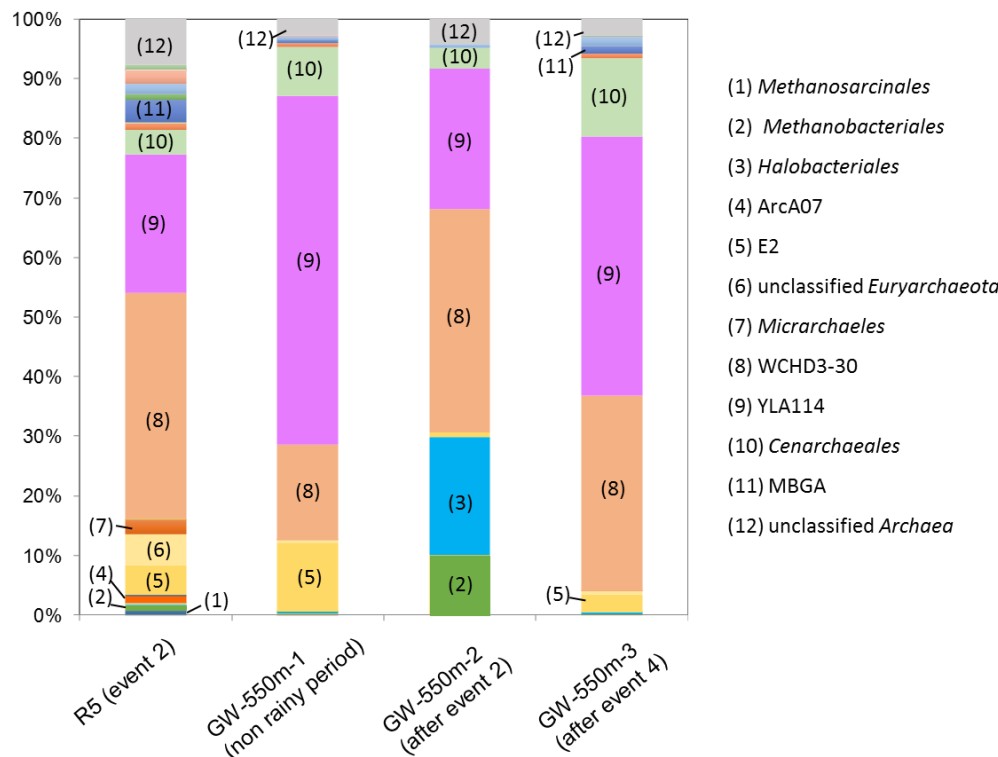

Figure 5. Relative abundance of archaeal orders in each events by next-generation sequencing analysis.



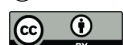

Table 1. Study site and observed environmental parameters.

| Site ID | Site name | Type of water | Altitude (m a.s.l.) | Sampling depth (m) | Observation period | Number of Investigation | Water temperature (°C) | pH | EC (μS cm⁻¹) | Eh (Pt) (mV) | DO, degree of saturation (%) |
|---------|-----------|---------------|---------------------|--------------------|--------------------|-------------------------|------------------------|----|-------------|--------------|------------------------------|
| SP-0 m | Shibakawa | Spring water | 726 | | 2012/6/15~2014/11/20 | n=19 | 10.1~11.9 | 5.84~7.35 | 49.5~128.0 | 343~497 | 86.7~92.9 |
| GW-42 m | Yodoshi | Groundwater | 150 | 42 | 2013/7/2~2014/11/20 | n=13 | 13.8~20.6 | 6.34~7.24 | 109.8~161.5 | 300~478 | 93.6~100.7 |
| GW-550 m | Aoki | Groundwater | 175 | 550* | 2013/7/2~2014/11/20 | n=13 | 13.7~19.8 | 6.63~8.26 | 122.7~142.8 | 246~447 | 73.0~89.9 |
| R1 | Go-gome | Rainwater | 2,364 | | 2013/6/17 | n=1 | 22.6 | 5.30 | - | - | - |
| R2 | Kokuyurin | Rainwater | 1,431 | | 2013/8/6~2014/10/16 | n=8 | 11.9~22.8 | 4.04~6.26 | - | - | - |
| R3 | Ni-gome | Rainwater | 1,081 | | 2013/6/17~2014/10/16 | n=11 | 14.2~23.4 | 3.94~5.82 | - | - | - |
| R4 | Asagiri | Rainwater | 850 | | 2013/6/17~2014/10/16 | n=15 | 13.0~31.9 | 4.12~6.14 | - | - | - |
| R5 | Shibakawa | Rainwater | 723 | | 2012/10/18~2014/10/16 | n=16 | 10.1~30.8 | 4.06~6.29 | - | - | - |

* Measurements were conducted for the water soon after it was pumped up.