# Peer review of "Tracking the direct impact of rainfall on groundwater at Mt. Fuji by multiple analyses including microbial DNA"

_Biogeosciences, 2017_

## Referee Comment (RC1) · Anonymous Referee #1 · 4 Oct 2017

Review of "Tracking the direct impact of rainfall on groundwater at Mt. Fuji byãĂĂmultiple analyses including microbial DNA" by Sugiyama and others

General comment: In this manuscript, the authors were trying to state that the information from microbial DNA in groundwaters was useful as a tracer to determine the contributions of runoff components. Presented data and descriptions include interesting and important information in groundwater pathways in the volcanic environments. However, there are several points have to be improved before publication in Biogeosciences.

1. For the essential part of discussions in this manuscript, it has been assumed that

the sources of transported bacteria were mainly situated in the soil horizons, and the sources of archaea were mainly in the "geologic layer". These assumptions may be common recognitions for general microbiologists. But, I feel there is a necessity to show evidences for guaranteeing these assumptions. Or, at least the authors have to explain how these assumptions were likely in this study site. 2. In "Introduction", the authors are telling:

"Whereas stable isotopic and 25 chemical analyses show average values of the water originated from various sources, microbes transported by groundwater suggest the route and place where they proliferated through their eco-physiological characteristics constrained by their optimal growth condition."

If the source locations (distributions) of each microbe could specified, pathways and origins of specific water sources could be identified. If the habitat of a microbe expanded in large spatial area, specifying capability of this microbe were low. Generally, this tendency can be applied also to isotope and chemical tracers. There is another issue. Conservativeness is also important for tracers. If you need to estimate relative contributions precisely of multiple end members, all tracers have to be conservative. In this point, microbial DNA may have disadvantage, because they may proliferate not only at the source points (area), but also in the pathways toward destinations. I think that the microbial DNA is a certainly useful tracer, but it can show its high capability being accompanied with other multiple tracers, such as isotopes and chemical tracers. The logic behind the above sentences was exaggerating the capability of microbial DNA as a tracer, if the authors can not show the sufficient evidences or generally accepted recognitions on the characteristics of microbial DNA as a tracers (spatially specific source and conservativeness).

Individual comments: P2, L2-5ïijŽ

"Though runoff process of stream water and runoff peak response time of streams influenced by rainfall have been well studied (e.g., Hubert et al., 1969; Onda et al.,

1999; Asai et al., 2001; Tekleab et al., 2014), runoff processes of groundwater affected directly by rainfall is not precisely explained."

Cited references were not always representative literatures for stating L1-2. For example, Dunne and Black (1970), Beven and Kirkby (1979), Burns et al. (2001), etc. many fundamental studies should be cited.

Dunne, T., and R. D. Black (1970), Partial area contributions to storm runoff in a small New England watershed, Water Resour. Res., 6(5), 1296– 1311, doi:10.1029/WR006i005p01296.

Beven, K. J., and M. J. Kirkby (1979), A physically based, variable contributing area model of basin hydrology, Hydrol. Sci. Bull., 24(1), 43– 69.

Burns, D. A., J. J. McDonnell, R. P. Hooper, N. E. Peters, J. E. Freer, C. Kendall, and K. J. Beven (2001), Quantifying contributions to storm runoff through end-member analysis and hydrologic measurements at the Panola Mountain Research Watershed (Georgia, USA), Hydrol. Processes, 15(10), 1903– 1924, doi:10.1002/hyp.246.

The statement of this sentence was not true. Many hydrological studies explained runoff processes of groundwater affected by rainfall.

e.g. McDonnell JJ, Bonell M, Stewart MK, Pearce AJ. (1990), Deuterium variations in storm rainfall: Implications for stream hydrograph separation. Water Resources Research. 26(3):455-8.

Kendall, C. and McDonnell, JJ (1993), Effect of intrastorm isotopic heterogeneities of rainfall, soil water, and groundwater on runoff modeling. IAHS Publication, 215, 41-48.

Figure 1: Why the unit of depth in the legend panel was m-1?

The line 3 – 5 of the caption was not formed a complete sentence. No indication for "Shibukawa" and no mark for "SP-0m" in the map.

Table 1: Is it possible to show the summary of isotope measurements?

[Figure]

---

## Author Comment (AC1) · 27 Oct 2017

Reply to Anonymous Referee #1

Review of "Tracking the direct impact of rainfall on groundwater at Mt. Fuji by Multiple analyses including microbial DNA" by Sugiyama and others General comment: In this manuscript, the authors were trying to state that the information from microbial DNA in groundwaters was useful as a tracer to determine the contributions of runoff components. Presented data and descriptions include interesting and important information in groundwater pathways in the volcanic environments. However, there are several points have to be improved before publication in Biogeosciences.

[Figure]

- - - - - - - - - -

Comment 1 (General): For the essential part of discussions in this manuscript, it has been assumed thatthe sources of transported bacteria were mainly situated in the soil horizons, and the sources of archaea were mainly in the "geologic layer". These assumptions may be common recognitions for general microbiologists. But, I feel there is a necessity to show evidences for guaranteeing these assumptions. Or, at least the authors have to explain how these assumptions were likely in this study site.

Reply 1: Thank you for the comment on the important standpoint of microbial distribution in subsurface environment. At first, the key point in consideration of microbes as an indicator of the route of groundwater is due to their "vertical" distribution. Sift of environment from soil to rock, slightly aerobic to absolute anaerobic, and increase in temperature with 3-4 °C/100m are the great constrain to characterize microbes in subsurface environment. Soil is thus clearly characterized from beneath environment with it very high abundance (108-9 cells/g; Katsuyama et al. 2008) and some dominant species as Burkholderiales and Bdellovibrionales (Garrity et al., 2005b). In order to clear the content, we add some words as follows;

p8, L26 [Original] An apparent predominance in the bacterial community of Burkholderiales suggests incorporation of microbes from soil...

[Revised] An apparent predominance in the bacterial community of Burkholderiales with high density suggests incorporation of microbes from soil...

p9,L2 [Original] Such extraction might increase the relative abundance of Bdellovibrionales in groundwater,

[Revised] Such extraction might increase the relative abundance of Bdellovibrionales including a typical soil-dweller as Peredibacter starrii in groundwater (Davidov and Jurkevitch, 2004),

Concerning Archaea, similarly we change wording;

p.9, L 19; [Original]It has been shown that archaeal abundance increased with depth in both terrestrial (Kato et al., 2009)

[Revised] Increasing in abundance of such archaea can be supported by the finding that archaeal abundance increased with depth in both terrestrial (Kato et al., 2009)

[References] Katsuyama et al. Denitrification activity and relevant bacteria revealed by nitrite reductase gene fragments in soil of temperate mixed forest. Microbes and Environments, 23:337-345, 2008.

Davidov and Jurkevitch, Diversity and evolution of Bdellovibrio-and-like organisms (BA-LOs), reclassification of Bacteriovorax starrii as Peredibacter starrii gen. nov., comb. nov., and description of the Bacteriovorax-Peredibacter clade as Bacteriovoracaceae fam. nov. Int J Syst Evol Microbiol. 54:1439-52, 2004.

- - - - - - - - - - -

Comment 2 (General): In "Introduction", the authors are telling: "Whereas stable isotopic and 25 chemical analyses show average values of the water originated from various sources, microbes transported by groundwater suggest the route and place where they proliferated through their eco-physiological characteristics constrained by their optimal growth condition."

If the source locations (distributions) of each microbe could specified, pathways and origins of specific water sources could be identified. If the habitat of a microbe expanded in large spatial area, specifying capability of this microbe were low. Generally, this tendency can be applied also to isotope and chemical tracers. There is another issue. Conservativeness is also important for tracers. If you need to estimate relative contributions precisely of multiple end members, all tracers have to be conservative. In this point, microbial DNA may have disadvantage, because they may proliferate not only at the source points (area), but also in the pathways toward destinations. I think that the microbial DNA is a certainly useful tracer, but it can show its high capability

being accompanied with other multiple tracers, such as isotopes and chemical tracers. The logic behind the above sentences was exaggerating the capability of microbial DNA as a tracer, if the authors cannot show the sufficient evidences or generally accepted recognitions on the characteristics of microbial DNA as a tracers (spatially specific source and conservativeness).

Reply 2: Thank you for your comment on the critical point. The great advantage of microbes as a tracer is stemmed from the fact that whether microbes which could suggest specific environment exist or not. Then, their relative abundance leads further discussion. In addition, the growth rate expressed by frequency of dividing cells (FDC in a given community) of subsurface microbes observed for groundwater and spring water in Mt. Fuji was very low (from 0.05 to 0.3 %, unpublished data) compared with surface waters (3 to 6 %). This suggests influence of proliferation of miscellaneous microbes through the pass of groundwater until examined may not alter the understanding shown here. This is shown in p9, Line32 as;

There was a question whether in situ population change through the growth in groundwater could be explained by their estimated growth rates. The doubling time of prokaryotes in the groundwater was estimated at 85 days, from the observed frequency of dividing cells' to the entire population (Newell and Christian, 1981). Thus, the possibility of altered populations via growth within a few weeks may be small. Microbes observed in the groundwater may represent the original locations where they grew.

- - - - - - - - - - -

Individual comments:

- - - - - - - - - - -

Comment 3 (Individual): P2, L2-5ïïjŽ "Though runoff process of stream water and runoff peak response time of streams influenced by rainfall have been well studied (e.g., Hubert et al., 1969; Onda et al., 1999; Asai et al., 2001; Tekleab et al., 2014), runoff

processes of groundwater affected directly by rainfall is not precisely explained." Cited references were not always representative literatures for stating L1-2. For example, Dunne and Black (1970), Beven and Kirkby (1979), Burns et al. (2001), etc. many fundamental studies should be cited.

Dunne, T., and R. D. Black (1970), Partial area contributions to storm runoff in a small New England watershed, Water Resour. Res., 6(5), 1296– 1311, doi:10.1029/WR006i005p01296. Beven, K. J., and M. J. Kirkby (1979), A physically based, variable contributing area model of basin hydrology, Hydrol. Sci. Bull., 24(1), 43– 69.

Burns, D. A., J. J. McDonnell, R. P. Hooper, N. E. Peters, J. E. Freer, C. Kendall, and K. J. Beven (2001), Quantifying contributions to storm runoff through end-member analysis and hydrologic measurements at the Panola Mountain Research Watershed (Georgia, USA), Hydrol. Processes, 15(10), 1903– 1924, doi:10.1002/hyp.246.

The statement of this sentence was not true. Many hydrological studies explained runoff processes of groundwater affected by rainfall.

e.g. McDonnell JJ, Bonell M, Stewart MK, Pearce AJ. (1990), Deuterium variations in storm rainfall: Implications for stream hydrograph separation. Water Resources Research. 26(3):455-8.

Kendall, C. and McDonnell, JJ (1993), Effect of intrastorm isotopic heterogeneities of rainfall, soil water, and groundwater on runoff modeling. IAHS Publication, 215, 41-48.

Reply 3: Thank you for your comment on the basic references. We change the sentence and add some references accordingly.

[Original] Though runoff process of stream water and runoff peak response time of streams influenced by rainfall have been well studied (e.g., Hubert et al., 1969; Onda et al., 1999; Asai et al., 2001; Tekleab et al., 2014), runoff processes of groundwater affected directly by rainfall is not precisely explained.

[Revised] Many hydrological studies explained runoff processes of groundwater affected by rainfall (e.g. Dunne and Black, 1970; McDonell et al., 1990; Beven et al., 2001; Tekleab et al., 2014). However, runoff process of groundwater affected directly by rainfall is not precisely explained.

- - - - - - - - - - -

Comment 4 (Individual): Figure 1: Why the unit of depth in the legend panel was m-1?

Reply 4: Yes, it was mistake. We correct the word in legend panel m-1 to m.

- - - - - - - - - - -

Comment 5 (Individual): The line 3 – 5 of the caption was not formed a complete sentence. No indication for "Shibukawa" and no mark for "SP-0m" in the map.

Reply 5: Thank you for your suggestion. We revised the map to show the site SP-0m. And the figure legend is revised as follows;

Figure 1. Study sites in western foot of Mt. Fuji. Red arrows indicate main fast flow (GETFLOWS; Kato et al., 2015 partially modified). Precipitation was sampled at R1 to R5. Groundwater was sampled at SP-0m, GW-42m and GW-550m. R1 is located at 2,364 m a.s.l., R2 is at 1,431 m a.s.l., R3 is at 1,081 m a.s.l., R4 is at 850 m a.s.l. and R5 is at 723 m a.s.l. SP-0m, spring water, shows sampling site of Shibakawa located at 726 m a.s.l. GW-42m, shallow well water obtained from 42 m, is located at Yodoshi with 150 m a.s.l. GW-550m, deep well water obtained from 550 m, is located at Aoki with 175 m a.s.l..

* Amount of precipitation for the studied area was recorded at Shiraito-no-taki Station of Japan Weather Association.

- - - - - - - - - - -

Comment 6 (Individual): Table 1: Is it possible to show the summary of isotope measurements?

Reply 6: We add the summary of isotopic data in Table 1.

[Figure]

[revised manuscript text omitted]
 | Adii | Groundwater | 175 | 550* | 2013/7/2~2014/11/20 | n=13 | 13.7~19.8 | 6.63~8.26 | 122.7~142.8 | 246~447 | 73.0~89.9 | -11.36~-10.81 | -76.1~-74.2 |
| R1 | Gogorne | Rainwater | 2,364 | | 2013/6/17 | n=1 | 22.6 | 5.30 | | | | -4.16 | -38.6 |
| R2 | Kofagorin | Rainwater | 1,431 | | 2013/8/9~2014/10/16 | n=8 | 11.9~22.8 | 4.04~6.26 | | | | -13.27~-7.01 | -95.2~-47.7 |
| R3 | Nagome | Rainwater | 1,081 | | 2013/6/17~2014/10/16 | n=11 | 14.2~23.4 | 3.94~5.82 | | | | -13.98~-4.61 | -89.1~-29.8 |
| R4 | Anagii | Rainwater | 850 | | 2013/6/17~2014/10/16 | n=15 | 13.0~19.9 | 4.12~6.14 | | | | -12.93~-3.62 | -92.4~-13.4 |
| R5 | Shibakawa | Rainwater | 723 | | 2012/10/18~2014/10/16 | n=16 | 10.1~19.8 | 4.06~6.29 | | | | -10.93~-3.09 | -80.1~-29.6 |

*Measurements were conducted for the water soon after it was pumped up.

20

**Fig. 5.** Table1

---

## Referee Comment (RC2) · Anonymous Referee #2 · 7 Nov 2017

General comments:

In this manuscript, the authors assess the impact of heavy rainfall events on Mt Fuji groundwater using isotopic, chemical and microbiological (DNA-based) tracers. The overall study yielded interesting and relevant results both from the chemical and the microbiological sides about the hydrology and the subsurface diversity of a unique site. However, the authors are making many important assumptions based on the microbial DNA analysis which are not necessarily true. The manuscript can be improved by nuancing the assumptions made and by the addition references on previous similar works in the introduction and the discussion sections. Besides this, the manuscript can

be published in Biogeosciences.

Specific comments:

-The introduction doesn't refer enough to previous microbiology works made on similar environments, to cite a few: ex. Zhou et al., 2012, Nyyssönen et al., 2013 for somewhat similar sites; Ben Maamar et al., 2015 for using a similar approach. The authors use too much space to justify their approach and not enough for referencing literature.

-I didn't find any substantial justification about the choice of using a piston-flow model rather another one like the Exponential piston model, except the occurrence of Archaea in the deep groundwater. Maybe adding some comments/schema on the geometry of the aquifer can help.

-Finding thermophilic microbes in environments with temperatures < 40°C is very common, same for halophilic microbes that can be found in low salts environments. Halobacteriales can be found in salted lakes, oceans and also, though not in high abundance, in temperate regions soils as well as on tree leaves, same for Methanobacteriales. In addition making some assumptions on microbes physiological optima using the classification at the order level is very risky and questionable. The authors should discuss the relative ubiquity of these microorganisms in different environments and maybe should specify the genus of these Archaea in order to give more credit to their assumptions. However, I strongly encourage the authors to moderate their assumptions based on detected taxa given the very low Archeae abundances observed.

-In Material and Methods, in the DNA extraction section no sampling triplicates were mentionned. Did the authors assessed the biological variability of their observations? If not, the authors should justify why and how their data might be representative of their environment.

-It would also be nice to add any water table measurements somewhere for each sampling campaigns as it may be relevant to discuss any increase/decrease in bacterial

density during rainfall events.

-The authors should also add the standard deviation for each total cell counts, as it helps to realize if observed increases in cells density are substantial, and gives an idea to readers of of the counting method sensitivity.

-most microbes in aquifers are living in an attached mode within biofilms, the authors should include a point in their discussion about how representative is a groundwater sample of the groundwater and subsurface biodiversity over time and space (specifically regarding the major attached fraction of microbes, see Flynn et al., 2008) and how it can affect their measurements.

-page 9 line 21, the reported Na+ concentrations are not particularly high compared to other aquifers (ex. Ben Maamar et al., 2015), specifically regarding Halobacteriales which are usually found in water saturated or nearly saturated with salt. They can live in somewhat less concentrated salt water though. Halobacteriales are mostly aerobes and they need organic material available which are usually in very low concentration in deep groundwater. The authors should add some information on the organic carbon availability in deep groundwater or maybe consider these Halobacteriales could also be introduced from soil.

-The paper would be improved with the addition of informations about the connectivity of the deep groundwater with surface, and if some surficial water inputs into deep groundwater are possible and in which proportions.

-The authors are a bit overselling the use of DNA as a flowpath tracer. Despite using DNA as a tracer is useful, it has several limits. For instance, microbes in aquifers are majorly living into heterogeneous biofilms and while some biofilms can be widespread, some others might develop only very locally and in very specific conditions. Defining the original location of each microbe based on their taxonomic assignation is far from being straightforward. Also, the authors should take into account that DNA can be more or less degraded according to the environmental conditions and keep in mind that the

vast majority of microbes are ubiquists, the main variable being their abundance in different environments. The use of DNA as a tracer is highly informative as long as used in combination with other tracers such as isotopic and chemical tracers.

-At the end of the discussion, unless I misunderstood it seems the authors assume the microbial diversity should go back to its initial structure after heavy rainfall events. This might be the case for very deep groundwater which seems to be poorly impacted by heavy rainfall but not necessary true for shallow groundwater that may host very fluctuating microbial diversity and structure over time because of the rapid water flow and variable contribution of soil over time.

Comments on figures:

Figure 4: Too many orders are represented, particularly for SP-0m-1. Please only show discussed or most relevant orders, or only depict orders representing more than 2 or 5 percents in relative abundance. Also please remove the shadow on colors.

Fig. S1, please add a table showing representative raw chemical concentrations for the different chemical species depicted, for comparison with other aquifers.

Technical corrections:

-page 7 line 16: what do 384, 268 and 278 correspond to? number of orders? Please reformulate -page 9 line 4: replace "was" by "were" -page 9 lines 4-7 this is a run-on sentence please split it into 2, and please clarify the point as this is not clear.

References:

Zhou, Y., Kellermann, C. & Griebler, C. Spatio-temporal patterns of microbial communities in a hydrologically dynamic pristine aquifer. FEMS Microbiol. Ecol. 81, 230–42 (2012).

Nyyssönen, M. et al. Taxonomically and functionally diverse microbial communities in deep crystalline rocks of the Fennoscandian shield. ISME J. 1–13 (2013).

doi:10.1038/ismej.2013.125

Flynn, T. M., Sanford, R. A. & Bethke, C. M. Attached and suspended microbial communities in a pristine confined aquifer. Water Resour. Res. 44, 1–7 (2008).

Ben Maamar, S. et al. Groundwater isolation governs chemistry and microbial community structure along hydrologic flowpaths. Front. Microbiol. 6, 1–13 (2015).

---

## Author Comment (AC2) · 29 Nov 2017

Reply to Anonymous Referee #2

General comments:

In this manuscript, the authors assess the impact of heavy rainfall events on Mt Fuji groundwater using isotopic, chemical and microbiological (DNA-based) tracers. The overall study yielded interesting and relevant results both from the chemical and the microbiological sides about the hydrology and the subsurface diversity of a unique site. However, the authors are making many important assumptions based on the microbial

DNA analysis which are not necessarily true. The manuscript can be improved by nuancing the assumptions made and by the addition references on previous similar works in the introduction and the discussion sections. Besides this, the manuscript can be published in Biogeosciences.

Reply:

Thank you so much for your precise reading our manuscript and valuable suggestions. We missed to refer a couple of suggested references, which we incorporate into the text. Detailed is shown below.

——

Specific comment 1:

The introduction doesn't refer enough to previous microbiology works made on similar environments, to cite a few: ex. Zhou et al., 2012, Nyyssönen et al., 2013 for somewhat similar sites; Ben Maamar et al., 2015 for using a similar approach. The authors use too much space to justify their approach and not enough for referencing literature.

Reply 1:

We agree with your suggestion and revise the text as follows;

[Original text]

To get indication on the route of groundwater we herein newly applied microbial DNA analysis focusing on heavy rainfall at the foot of Mt. Fuji located in central Japan, which is the largest Quaternary stratovolcano in Japan with a peak at 3,776 m a.s.l. At the foot of this mountain we previously found that pH of groundwater decreased from 7.29 to 7.02 a few weeks after a typhoon in August and September 2011 (total rainfall was > 800 mm) (Segawa et al., 2015) at 200 m a.s.l. (page 2 lines 6-8ïïjĽ

[Revised text]

Development in gene sequence of in situ microbial community enables us to discuss relation between environment and community constituents (Zhou et al. 2012, Nyyssinen et al. 2014). And, population dynamics of predominant prokaryote can be discussed with changes in environment. Concerning subsurface environment Ben Maamar et al. (2015) recently showed a good correlation between different condition of groundwater with oxygen and dominant microbial population, and suggested mixing of groundwater. We herein tried to apply microbial DNA analysis to indicate the route of groundwater focusing on heavy rainfall at the foot of Mt. Fuji located in central Japan, which is the largest Quaternary stratovolcano in Japan with a peak at 3,776 m a.s.l. (page 2 lines 6-12ïijĽ

——

Specific comment 2:

I didn't find any substantial justification about the choice of using a piston-flow model rather another one like the Exponential piston model, except the occurrence of Archaea in the deep groundwater. Maybe adding some comments/schema on the geometry of the aquifer can help.

Reply 2:

Thank you for the comment. We discuss the route of groundwater based on apparent age and the apparent age assumes piston flow of the groundwater. The preceding study performed in this study area applied the piston flow model (Tosaki et al., 2011). In addition, some studies on the groundwater age conducted in volcanic area applied piston flow to get apparent age (e.g., Koh et al., 2007). Thus, we discuss the influence of heavy rainfall appeared in deep groundwater by the concept of piston flow. And, to discuss the flow system isn't the aim herein. We just intend to suggest the impact of heavy rainfall on the deeper groundwater from actual increase in abundance of archaea and clear difference found in archaeal constituents. Water doesn't tell us any influence of heavy rainfall there, but particles.

[Reference]

Tosaki et al., Estimation of groundwater residence time using the 36Cl bomb pulse, Ground Water, 49, 891-902, 2011.

Koh, et al., Evidence for terrigenic SF6 in groundwater from basaltic aquifers, Jeju Island, Korea: Implications for groundwater dating, J. Hydrol., 339, 93-104, 2007.

—–

Specific comment 3:

Finding thermophilic microbes in environments with temperatures < 40°C is very common, same for halophilic microbes that can be found in low salts environments. Halobacteriales can be found in salted lakes, oceans and also, though not in high abundance, in temperate regions soils as well as on tree leaves, same for Methanobacteriales. In addition making some assumptions on microbes physiological optima using the classification at the order level is very risky and questionable. The authors should discuss the relative ubiquity of these microorganisms in different environments and maybe should specify the genus of these Archaea in order to give more credit to their assumptions. However, I strongly encourage the authors to moderate their assumptions based on detected taxa given the very low Archeae abundances observed.

Reply 3:

Thank you for your comment on how to interpret the meaning of the findings of some certain group of bacteria. The point we insist herein is from where they came. If you find thermophilic prokaryote from cold water, which leads a question as from where do they come? Fig.5 expressed contribution of each group of archaea at the level of Order, but the original data, Haloarcula cpmprised 99.7 % of Halobacteriales and Methanothermobacter comprised 97.4 % of Methenobacteriales. Thus, we add this information into the text; And we think that an increasing in contribution of both constituents of Halobacteriales (Haloarcula) and Methenobacteriales (Methanothermobacter) is plausible because they might be retrieved from deep subsurface environment as unique constituents which were just found after the heavy rainfall when the density of archaea sharply increased after the Event 2 (Fig. 3).

[Original text]

However, we observed an interesting increase in abundance of Archaea at GW-550m 2 weeks after event 2, which was supported by an apparent change in constituents of archaeal OTUs. Halobacteriales, which inhabit environments with high concentrations of sodium and Methanobacteriales, a strict anoxic methane producer, were dominant members after the torrential rainfall. (page 9 lines 16-19)

[Revised text]

However, we observed an interesting increase in abundance of Archaea at GW-550m 2 weeks after event 2, which was supported by an apparent change in constituents of archaeal OTUs. Halobacteriales comprised of Haloarcula with 99.7%, which inhabit environments with high concentrations of sodium and Methanobacteriales comprised of Methanothermobacter with 97.4%, a strict anoxic methane producer, were dominant members after the torrential rainfall. (page 9 lines 16-19)

——

Specific comment 4:

In Material and Methods, in the DNA extraction section no sampling triplicates were mentioned. Did the authors assessed the biological variability of their observations? If not, the authors should justify why and how their data might be representative of their environment.

Reply 4:

We concentrated 10 L of groundwater for each analysis, and it is almost practical limit to do in situ environment for each observation (We reserved far more number of sam-

ples.). Though we did not get water with triplicate for each sample, e.g., the similarity in the first three bands of DGGE pattern obtained from before the Event-2 at SP-0m (Supplement Fig. S2 (original)) supports that the employed method might not be biased much to represent the microbial community so far examined for the groundwater.

——

Specific comment 5:

It would also be nice to add any water table measurements somewhere for each sampling campaigns as It may be relevant to discuss any increase/decrease in bacterial density during rainfall events.

Reply 5:

Thank you for your suggestion. We add a figure of fluctuation of the amount of discharge of spring water at SP-0m as Figure S1. Figure S1 shows the amount of discharge observed at SP-0m did not affect the density of microbes. Sharply increased discharge observed during Event 4 correlated with just the density of Archaea in deep groundwater at GW-550m, which well located 1.2km downstream from the site of SP-0m.

——

Specific comment 6:

The authors should also add the standard deviation for each total cell counts, as it helps to realize if observed increases in cells density are substantial, and gives an idea to readers of of the counting method sensitivity.

Reply 6:

We add standard deviation to Figure 3 calculated from the total density of prokaryotes and its individual contribution of Bacteria and Archaea. Bacterial density of June 17 2013 was corrected in the revised Figure 3. The previous Figure 3 (c) showed observation date incorrect (each point is shown one time ahead). This is also revised in the new Figure 3 (c).

———

Specific comment 7:

most microbes in aquifers are living in an attached mode within biofilms, the authors should include a point in their discussion about how representative is a groundwater sample of the groundwater and subsurface biodiversity over time and space (specifically regarding the major attached fraction of microbes, see Flynn et al., 2008) and how it can affect their measurements.

Reply 7:

Yes, we agree on the concept that abundance of microbial particles exceeds in attached form in subsurface environment. An apparent increase in relative contribution of Methannobacteriaales and Holobacteriales shown in Fig. 5 for the deep groundwater after the heavy rainfall, thus, can be ascribable to detached from the rocks in the deeper layer, which was suggested from the chemistry. Similar estimate is applied to explain the sharp increase in Bacterial abundance in spring water at SP-0m (Fig. 3, (a)), which was mostly supplied from soil constituents through the extraction after Event 2 (page 8 line 25 – page 9 line 3). To make more clear the discussion, ï¡Ůe add a reference you suggested into the part of Discussion as follows. Thank you.

[Revised text]

Thus, not only strict anaerobic but halophilic archaea may be abundant within the deep subsurface environment of the study area, although they were not retrieved from groundwater in other examinations, likely because they were embedded in the matrix of geologic layers (Flynm et al. 2008). (page 9 lines23-26ïijĽ

———

Specific comment 8:

page 9 line 21, the reported Na+ concentrations are not particularly high compared to other aquifers (ex. Ben Maamar et al., 2015), specifically regarding Halobacteriales which are usually found in water saturated or nearly saturated with salt. They can live in somewhat less concentrated salt water though. Halobacteriales are mostly aerobes and they need organic material available which are usually in very low concentration in deep groundwater. The authors should add some information on the organic carbon availability in deep groundwater or maybe consider these Halobacteriales could also be introduced from soil.

Reply 8:

Yes, we just refer the concentration of Na+ obtained from the examined deep water was higher than the other shallow groundwater in the examined area. The reviewer asked the possibility to retrieved Halobacteriales from soil. But, we found Methanother-mobacter, a strict anaerobe, together with them. This could suggest that the possibility to retrieve Halobacteriales from soil might not be high.

——

Specific comment 9:

The paper would be improved with the addition of informations about the connectivity of the deep groundwater with surface, and if some surficial water inputs into deep groundwater are possible and in which proportions.

Reply 9:

Thank you for your suggestion. But the diversity of microbes inhabit in surface environment in particular in soil, which can be regarded as the topmost, must be very high and varies with the characteristics of soil (e.g. Katsuyama et al. 2008). Thus we simply refer to the predominant prokaryote, Burkholderia, in major (Fig.4).

[Reference]

Katsuyama et al. Denitrification activity and relevant bacteria revealed by nitrite reductase gene fragments in soil of temperate mixed forest. Microbes and Environments, 23:337-345, 2008

——

Specific comment 10:

The authors are a bit overselling The use of DNA as a flowpath tracer. Despite using DNA as a tracer is useful, It has several limits. for instance, microbes in aquifers are majorly living into heterogeneous biofilms and while some biofilms can be widespread, some others might develop only very locally and in very specific conditions. Defining The original location of each microbe based on their taxonomic assignation is far from being straightforward. also, The authors should take into account that DNA can be more or less degraded according to The environmental conditions and keep in mind that The vast majority of microbes are ubiquists, The main variable being their abundance in different environments. The use of DNA as a tracer is highly informative as long as used in combination with other tracers such as isotopic and chemical tracers.

Reply 10:

We totally agree with your comment that "The use of DNA as a tracer is highly informative as long as used in combination with other tracers such as isotopic and chemical tracers." To get information on the degradation for each target prokaryote, in particular, is the subject to be studied in the next step. Thus the last sentence of Discussion is; In addition to the chemical analyses of groundwater, we showed that microbes could show the route of groundwater in the invisible subsurface environment. But, a sentence in "Conclusion" we modify as follows;

[Original text]

Here, we first indicated the route of groundwater using a next-generation sequencing

analysis of Bacteria and Archaea. (page 10 lines 21-22)

[Revised text]

Here, we first showed the possibility to chase the route of groundwater using a next-generation sequencing analysis of Bacteria and Archaea for the event of heavy rainfall. (page 10 lines 22-24)

——

Specific comment 11:

At the end of the discussion, unless I misunderstood it seems the authors assume the microbial diversity should go back to its initial structure after heavy rainfall events. This might be the case for very deep groundwater which seems to be poorly impacted by heavy rainfall but not necessary true for shallow groundwater that may host very fluctuating microbial diversity and structure over time because of the rapid water flow and variable contribution of soil over time.

Reply 11:

Thank you for your comment and we mostly agree with your suggestion. As you suggested, resilience in deep archaeal community constituents was shown herein to some extent (Fig. 5). But, we think we do not have enough information about the ability of resilience in subsurface microbial community. We just discussed a possible estimate on the influence of heavy rainfall even for deep groundwater from the finding just after the heavy rainfall.

——

Comments on figures 12:

Figure 4: Too many orders are represented, particularly for SP-0m-1. Please only show discussed or most relevant orders, or only depict orders representing more than 2 or 5 percents in relative abundance. Also please remove the shadow on colors.

Reply 12:

Thank you for your suggestions. We modify the legend of Figure 4 showing orders contributed exceeding 2%.

——

Comments on figures 13:

Fig. S1, please add a table showing representative raw chemical concentrations for the different chemical species depicted, for comparison with other aquifers.

Reply 13:

Thank you for your suggestion. We add Table S2 showing chemical character as averages of major ions.

——

Technical correction 14:

page 7 line 16: what do 384, 268 and 278 correspond to? number of orders? Please reformulate

Reply 14:

Yes, the numbers are amounts of order, so we change the sentence as follows;

[Original text]

Next-generation sequencing retrieved diversified community constituents at the level of order with 384, 268 and 278 from rainwater (R5), spring water before event 2 (SP-0m-1) and spring water after event 2 (SP-0m-2), respectively. (page 7 lines 16-17)

[Revised text]

Number of constituents retrieved by Next-generation sequencing at the level of Order was 384, 268 and 278 for rainwater (R5), spring water before event 2 (SP-0m-1) and

spring water after event 2 (SP-0m-2), respectively. (page 7 lines 16-17)

———

page 9 line 4: replace "was" by "were"

Reply 14-2:

We correct the word "was" to "were". ([Original text] page 9 line 4, [Revised text] page 9 line 5)

———

page 9 lines 4-7 this is a run-on sentence please split it into 2, and please clarify the point as this is not clear.

Reply 14-3:

We separate this sentence into 2 as followed.

[Original text]

Furthermore, sequences affiliated with thermophilic bacteria was scarcely retrieved from the samples of the examined SP-0m after event 2, which supports the assertion of enforced piston flow through a deep subsurface zone > 600 m which given temperature exceeding 40 °C, where thermophilic bacteria inhabited was not considerable. (page 9 lines 4-7)

[Revised text]

Furthermore, sequences affiliated with thermophilic bacteria were scarcely retrieved from the samples of the examined SP-0m where thermophilic bacteria inhabited was not considerable after event 2. This finding supports the assertion of enforced piston flow through a deep subsurface zone > 600 m which given temperature exceeding 40 °C. (page 9 lines 5-7)

———

[Figure]

[Figure]

Figure 3. Changes in community structure of prokaryotes in groundwater. (a) Groundwater at SP-0m, (b) shallow groundwater at GW-42M, (c) deep groundwater at GW-550m. Black and open arrows indicate the rainfall event; Event 1, Event 2, Event 3 and Event 4. Black arrows particularly indicate the torrential rainfall. Yellow arrows indicate the signature of impact of rainfall.

**Fig. 1.** Figure 3

[Figure]

Figure S1. Fluctuation of the Amount of discharge at the spring of SP-0m.

**Fig. 2.** Figure S1

Table S2. Chemical character of spring and groundwater.

(meq L⁻¹)

| Site ID | n | Na⁺ | K⁺ | Mg²⁺ | Ca²⁺ | Cl⁻ | NO₃⁻ | SO₄²⁻ | HCO₃⁻ |
|---------|-----|-------|-------|-------|-------|-------|-------|-------|-------|
| SP–0m | 15 | 0.258 | 0.032 | 0.200 | 0.429 | 0.115 | 0.057 | 0.151 | 0.670 |
| GW–42m | 13 | 0.313 | 0.046 | 0.308 | 0.578 | 0.086 | 0.122 | 0.104 | 1.002 |
| GW–550m | 13 | 0.631 | 0.015 | 0.177 | 0.353 | 0.088 | 0.007 | 0.243 | 0.818 |
| R1 | 1 | 0.002 | 0.002 | 0.001 | 0.025 | 0.002 | 0.004 | 0.000 | 0.059 |
| R2 | 8 | 0.034 | 0.002 | 0.013 | 0.011 | 0.040 | 0.019 | 0.024 | 0.053 |
| R3 | 11 | 0.034 | 0.002 | 0.013 | 0.011 | 0.040 | 0.019 | 0.024 | 0.035 |
| R4 | 15 | 0.011 | 0.012 | 0.006 | 0.008 | 0.013 | 0.026 | 0.058 | 0.021 |
| R5 | 15 | 0.017 | 0.006 | 0.010 | 0.013 | 0.020 | 0.021 | 0.026 | 0.032 |

**Fig. 3.** Table S2

[revised manuscript text omitted]